## Comment

 

**Subject Category:**
Biology (whole organism)

ecology/behaviour

marine mammals, ichnology, deep-diving mammals, Atacama Trench, sidescan sonar, towed camera sled

**Author for correspondence:**
Autun Purser
e-mail: autun.purser@awi.de

# Depression chains in seafloor of contrasting morphology, Atacama Trench margin: a comment on Marsh *et al.* (2018)

Autun Purser[1], Helena Herr[1,2], Simon Dreutter[1], Boris Dorschel[1], Ronnie N. Glud[3,4], Laura Hehemann[1], Ulrich Hoge[1], Alan J. Jamieson[5], Thomas D. Linley[5], Heather A. Stewart[6] and Frank Wenzhöfer[1,7]

[1]Alfred Wegener Helmholz Institute of Polar and Marine Research, Bremerhaven, Germany
[2]Center of Natural History, University of Hamburg, Hamburg, Germany
[3]Department of Biology, University of Southern Denmark, 5230 Odense, Denmark
[4]Department of Ocean and Environmental Sciences, Tokyo University of Marine Science and Technology, Tokyo, Japan
[5]School of Natural and Environmental Sciences, Newcastle University, Newcastle Upon Tyne, UK
[6]British Geological Survey, Lyell Centre, Edinburgh, UK
[7]Max Planck Institute for Marine Microbiology, Bremen, Germany

AP, 0000-0001-5427-0151; HH, 0000-0002-5028-2419;
SD, 0000-0002-0878-0780; BD, 0000-0002-3495-5927;
RNG, 0000-0002-7069-893X; AJJ, 0000-0001-9835-2909;
TDL, 0000-0002-6583-3105; HAS, 0000-0002-5590-6972;
FW, 0000-0002-4621-0586

This comment presents acoustic and visual data showing deep seafloor depression chains similar to those reported in Marsh *et al.* (*R. Soc. open sci.* 5: 180286), though from a different deep-sea setting. Marsh *et al.* present data collected during cruise JC120 from polymetallic nodule rich sites within the Clarion-Clipperton Fracture Zone (CCFZ), at water depths of between 3999 and 4258 m. Within this comment, we present data collected with equivalent acoustic and imaging devices on-board the RV Sonne (SO261—March/April 2018) from the Atacama Trench, approximately 4000 m depth, which shows comparable depression chains in the seafloor. In contrast with the CCFZ observations, our study area was wholly free of polymetallic nodules, an observation therefore weakening the 'ballast collection' by deep-sea diving mammals formation hypothesis discussed in their paper. We support their alternate hypothesis that if these features are indeed generated by deep-diving

megafauna, then they are more likely the resultant traces of infauna feeding or marks made during opportunistic capture of benthic fish/cephalopods. We observed these potential prey fauna with lander and towed camera systems during the cruise, with example images of these presented here. Both the SO261 and JC120 cruises employed high-resolution sidescan systems at deployment altitudes seldom used routinely until the last few years during scientific deep-sea surveys. Given that both cruises found these depression chains in contrasting physical regions of the East Pacific, they may have a more ubiquitous distribution than at just these sites. Thus, the impacts of cetacean foraging behaviour on deep seafloor communities, and the potential relevance of these prey sources to deep-diving species, should be considered.

# 1. Introduction and methodology

During March/April 2018, the research vessel RV Sonne investigated the Atacama Trench, offshore the west coast of South America (Cruise SO261), as part of the multidisciplinary HADES European Research Council (HADES-ERC) study of deep trench systems. A towed sled (the Ocean Floor Observation and Bathymetry System (OFOBS[1])) incorporating cameras (still and video) and sidescan sonar was deployed at an altitude of 1.5–2 m above seafloor at water depths between 3500 and 6000 m at seven locations (figure 1). The still camera and sidescan systems were similar to those used by Marsh *et al.* [2], though the additional mounting of a video camera and a 50 cm spaced tri-laser sizing system on OFOBS allowed georeferenced video frames to be extracted for subsequent creation of three-dimensional seafloor models [1]. In addition to conducting OFOBS tows, baited HD camera landers were also deployed, collecting data on bait-attending fish and mobile fauna present at various depths within the surveyed area.

# 2. Results

At 20°19′50″ S 71°0′20″ W on the eastern flank of the Atacama Trench, at depths of 3990–4140 m (cruise survey ID SO261/109-1, figure 1), a number of extended chains of depression features, reminiscent of those presented in Marsh *et al.* [2] were observed during an OFOBS survey of approximately 2.5 km length (figure 2 and electronic supplementary material, S1). By chance, one depression was passed directly over and imaged by still (figure 3) and video cameras (video frames mosaiced into figure 4). By using the tri-laser sizing system of the OFOBS and the PAPARA(ZZ)I 2.6 software [3], the imaged depression disturbance area was estimated to be $55 \pm 10$ cm in width and 1.5–2 m in length. In the concurrently collected sidescan sonar data, the average spacing of the depressions in the most extensive chain imaged (presented in figure 2) was determined to be approximately 8.5 m, roughly the length of a Cuvier's beaked whale. At one point a chain encountered a small ridge, the likely surface expression of a small fault (western section of figure 2), at which point the depression chain is offset by approximately 30 m before continuing in a roughly E–W direction. The sidescan data showed all depressions to be elongated in the direction of the depression chain. These depression chains weave across, in and out of the surveyed region, with at least one chain appearing to be in excess of 500 m length. A three-dimensional model of the imaged depression was generated using frames extracted from the concurrently collected video (electronic supplementary material, S2 and S3), from which an estimated maximum depression depth of approximately 15 cm was estimated. Further depressions were partially imaged by the camera systems at other locations (electronic supplementary material, S4).

In some sections of cruise survey SO261/109-1, secondary chains of depressions bisect longer chains (as shown in figure 2 and the eastern section of S1). Even though they interweave, the course of all chains observed was roughly perpendicular to the depth contours (figure 1). As in the CCFZ data, we observed variations in the sharpness of the outlines of the depression features present on the Atacama Trench flank. Two chains of depressions with very sharp, comparable outlines interweave each other at the eastern extremity of S1, with a further pair of chains observed toward the west of S1 (shown in detail in figure 1). The second of these chain pairs exhibited less distinct edges in the acoustic data. In the case of both these chain pairings, the spacing of the individual depressions, while uniform for a particular chain, differed from that of the companion chain by approximately 15%.

Both OFOBS and baited camera landers recorded a range of fish (primarily the macrourids *Coryphaenoides armatus* and *Coryphaenoides yaquinae*, though also the ophidiid *Bassozetus* sp. and an unidentified Ipnopidae) at approximately 4000 m depth, and other typical deep-sea benthic fauna including crustacean, ophiuroid, holothurian, jellyfish and hemichordate species. Several benthic

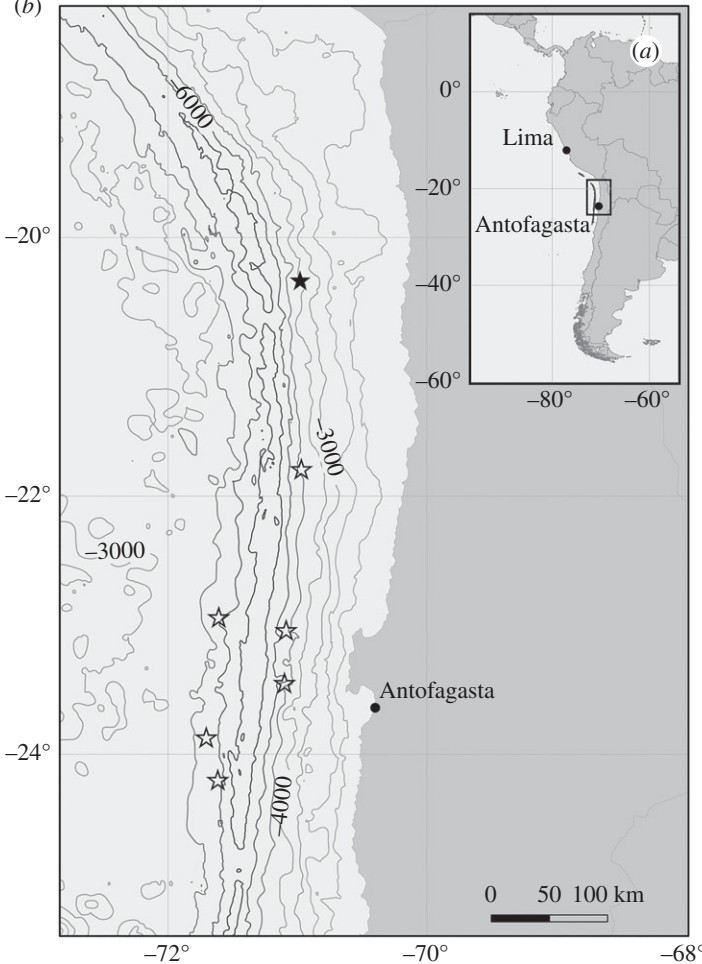

**Figure 1.** (*a*) Location of the Atacama Trench and region of study. (*b*) Map showing the location of OFOBS deployments conducted during SO261. Cruise survey SO261/109-1, with image and acoustic data collected and the focus of the current study, is indicated by a black star. Open star symbols represent OFOBS deployments where only image data was collected successfully.

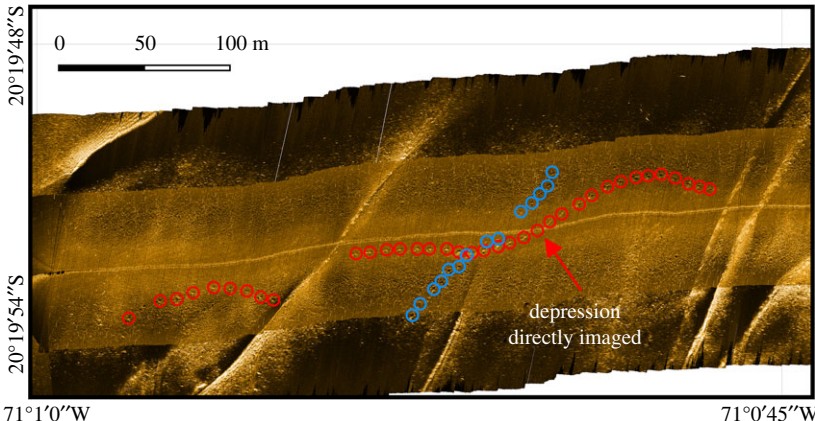

**Figure 2.** Georeferenced, processed sidescan data collected from a flight height of 1.5–2 m showing the chain of depressions reminiscent of those presented in Marsh *et al.* [2]. Individual depression features outlined by red and blue circles. Red arrow indicates the position of the image of seafloor given in figures 3 and 4.

incirrate octopi of unknown species were also imaged, reminiscent of those recently reported in Purser *et al.* [3] from comparable depth in the DISCOL region of the East Pacific. Additionally, indications of infauna activity were observed, such as burrows, mounds and signs of sediment disturbance by emergent fauna (figure 5).

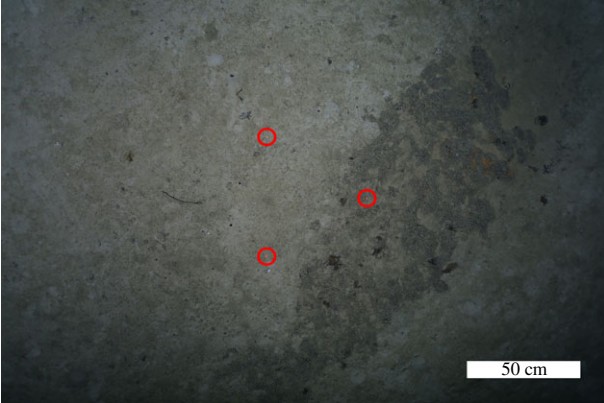

**Figure 3.** Image of depression feature taken from 1.5 m altitude. Lazer points (circled in red) have a 50 cm spacing.

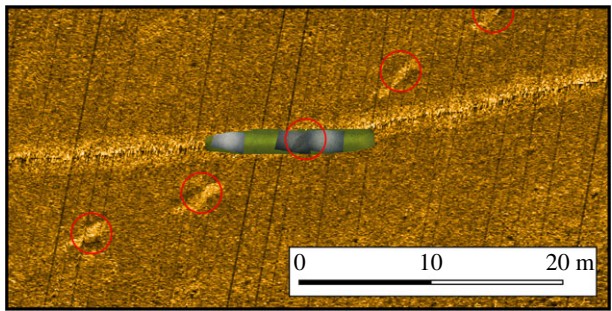

**Figure 4.** Two-dimensional georeferenced mosaic of OFOBS image frames (grey tint) and video frames (green tint) mapped directly onto sidescan derived bathymetric data.

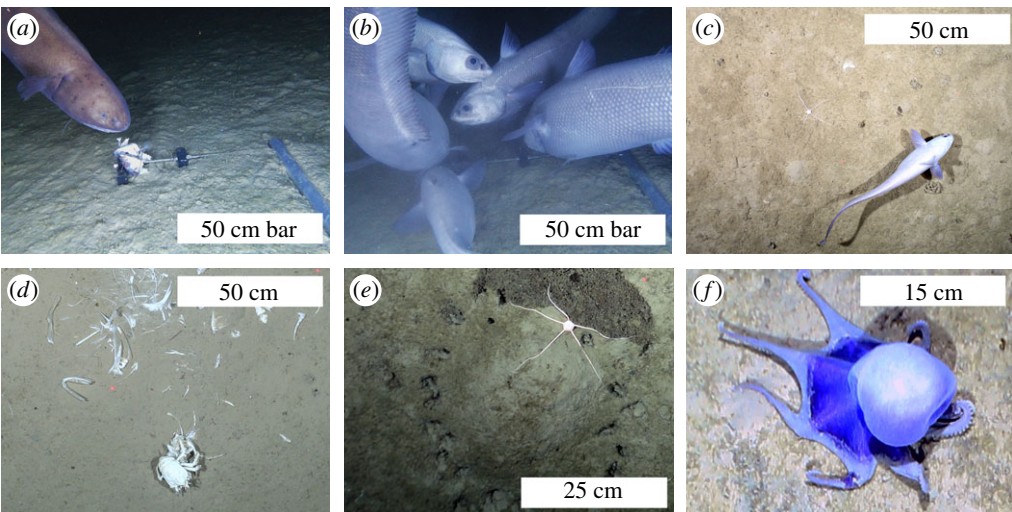

**Figure 5.** Mobile fauna observed at approximately 4000 m depth during SO261. (*a*) *Bassozetus* spp. (*b,c*) *Coryphaenoides* spp. (*d*) Crustacean on fish fall. (*e*) Ophiuroid on mound. (*f*) Unknown incirrate octopus.

# 3. Discussion

## 3.1. Depression geomorphology and distribution

The ultra-low deployment height of the sonar systems used by Marsh *et al.* [2] and ourselves seem to indicate that such depression chains in sediments of approximately 4000 m depth are potentially widespread geomorphological features (at least in the East Pacific) not detected by previous iterations

of deep-sea survey equipment. Though the chains were broadly similar in spacing, size and form to those presented in Marsh *et al.* [2] within the CCFZ, the seafloor characteristics differed greatly in hard substrate availability, with our surveyed area of the Atacama Trench wholly free of polymetallic nodules, which were abundant in the CCFZ.

## 3.2. Formation hypotheses

Like Marsh *et al.* [2], we find it difficult to assign an abiotic formation mechanism for these depression chains. However, unlike the CCFZ study area of Marsh *et al.*, there are potential mechanisms for fluid flow within the Atacama Trench setting with bend-related faulting within subduction trenches suggested to play a role in local fluid circulation [4]. Depressions, such as pockmarks related to fluid-flow processes, have diverse morphologies and can occur in both random and non-random distributions. Distribution of these crater-like depressions is controlled by underlying geological features such as faults or buried channels [5–7]. However, the depressions observed herein are contrary to the fault trend evident in these data (figure 2) suggesting that there is no underlying geological control related to the formation of these depression chains. We did not image any fish larger than 1 m, and the larger individuals we did observe were moving slowly above the seafloor (*Coryphanoides armatus* and *C. yaquinae*), not interacting with it in any notable way, on any occasion. No sediment excavation or disturbance by fish was apparent.

We believe the overriding hypothesis that cetaceans are causing these depressions is greatly strengthened by our new data. Cetacean contact with the seafloor and seafloor cable infrastructure has been reported (at least in shallower regions of our area of research) since the nineteenth century, and particularly prominently in the eastern Pacific [8]. Both deep-diving sperm whales (*Physeter macrocephalus*) and Cuvier's beaked whales (*Ziphius cavirostris*) are numerous in the region and eastern Pacific in general [9], Cuvier's beaked whales are preferentially observed in regions with seafloor slope, such as the Atacama Trench margin [10]. As highlighted in Marsh *et al.* [2], no whales have yet been observed (i.e. tagged) diving to the seafloor depths associated with CCFZ or the region of the Atacama Trench investigated here. The maximum dive depth recorded in cetaceans thus far has been 2992 m by a Cuvier's beaked whale [11]. These whales are probably physiologically capable of diving much deeper, as deep as 5000 m [12,13], and certainly spend extended periods close to the seafloor during foraging dives [14,15].

Marsh *et al.* [2] present a range of hypotheses as to why whales may be diving to abyssal depths and interacting with the seafloor; some of which are supported by our new observations, others weakened. The idea that the whales forming the depression chains observed on the eastern flank of the Atacama Trench were seeking stones to consume, to function in the role of gastroliths, is highly unlikely, given the total absence in images of any surface solid material in this survey. Further, marine vertebrates with hydrofoil limbs (such as penguins, otariid pinnipeds and extinct plesiosaurs) have been suggested to use such material, rather than caudal finned cetaceans ([15] and references therein), such as the negatively buoyant beaked whales [16]. We also believe it unlikely that the active pursuit of a particular fish or cephalopod individual would result in the depression patterns observed. Our depression tracks formed continuous, unbroken, undulating and evenly spaced chains, whereas active pursuit may be expected to result in tracks with sudden changes of direction or variations in swimming speed (and therefore depression spacing). The chains we observed were also unlike the chaotic and intense 'surface of the moon' depressions reported for shallow bottlenose dolphin (*Tursiops truncatus*) feeding [17] or the short chains and parallel splayed arrays of elongated depressions as formed by benthic feeding gray whales (*Eschrichtius robustus*) [18,19]. The individual outline dimensions of the depressions do match almost exactly those observed in Woodside *et al.* [20]; the authors suggested these to have been made by Cuvier's beaked whales in the Mediterranean. The depressions imaged during our study appear far less fresh, partially infilled with sediment and biodetritus, than the examples in [20] or indeed those in Marsh *et al.* [2]. We observed no clear central groove [20] in the depressions imaged on the Atacama Trench.

Of the deep-diving whale species reported in the eastern Pacific, sperm whales and Cuvier's beaked whales are the most widely reported, with information on diets for both species commonly published from opportunistic strandings from diverse and contrasting locations [21–24], possibly reflective of local dietary availabilities to individuals from these ubiquitous species rather than of firm dietary preferences. The wide range of fauna observed in the image data collected during our cruise include species occasionally found in the stomach contents of both Cuvier's beaked and sperm whales, although generally sperm whales seem to prefer deep-sea squid from higher in the water column [24,25], despite spending approximately 50 min very close to the seafloor during documented dives [14]. Images of deep squid were captured during deployment of the OFOBS instrument. Possibly opportunistic grazing of the benthic environment is made prior to a return to the surface, particularly if primary prey targets were not

encountered during the dive, due to either absence of prey of any acoustic disturbance. Additionally, younger or stressed individuals may graze more readily on slow-moving benthic fauna than on fast-moving free-swimming prey higher in the water column. Foraging for a mixed, varied diet of infauna and fauna, without the active pursuit of prey individuals [21,26], would likely result in the extended depression chains observed on the Atacama Trench margin. Such opportunistic feeding, when conducted in regions containing small stones or nodules, may result in accidental consumption, potentially accounting for the occasional individuals found with such inorganic material within their stomachs [27,28]. From our data, the interweaving of depressions of similar age characteristics would support a loose group/small pod of foraging whales. Such roughly parallel chains of depressions can also be seen in fig. 3*b* of Marsh *et al.* [2] in the wider spatial data collected by their AUV system. Potentially, these depression chains then are merely the individual components of a more spatially extensive seafloor interaction made by a pod of foraging whales.

Data accessibility. The datasets supporting this article have been uploaded as part of the electronic supplementary material. All additional georeferenced sidescan and image-based data collected via OFOBS during cruise survey SO261/109-1 are available from PANGAEA in full, native resolution at https://doi.pangaea.de/10.1594/PANGAEA.894733.

Authors' contributions. A.P., L.H., U.H. and F.W. collected the sidescan and image data. F.W. was principal investigator while R.N.G. was the scientific leader during SO261 and both conceived the study. S.D., B.D., L.H. and H.A.S. georeferenced and processed the geomorphological data. A.J.J. and T.L. collected and analysed the fish data. A.P. and H.H. analysed the combined dataset and prepared the manuscript. All authors actively contributed to the manuscript and gave their final approval for publication.

Competing interests. We have no competing interests.

Funding. The SO261 cruise fieldwork was funded by the HADES-ERC Advanced Grant ('Benthic diagenesis and microbiology of hadal trenches' Grant agreement No. 669947) awarded to R.N.G. (University Southern Denmark). Ship-time on board the RV SONNE was financed by BMBF and was awarded to F.W. (AWI/MPI) M. Zabel (MARUM), and R.N.G. (University of Southern Denmark/Tokyo University). Antje Boetius oversaw the development of the OFOBS system, which was funded by ERC advanced grant 'Abyss' (Investigator grant no. 294757) and the FRAM project.

Acknowledgements. We would like to thank the crew and scientific party of RV SONNE cruise SO261 for their able and enthusiastic assistance in the collection of the datasets discussed within this comment. H.A.S. publishes with the permission of the Executive Director of the British Geological Survey (United Kingdom Research and Innovation). Two anonymous reviewers are thanked for constructive review of the initial manuscript, input from which has improved this current draft.

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
