## [Reviewer comments · Royal Society Open Science]

Review History

RSOS-182053.R0 (Original submission)

Review form: Reviewer 1 (Sascha Hooker)

Is the manuscript scientifically sound in its present form?

Yes

Are the interpretations and conclusions justified by the results?

Yes

Is the language acceptable?

Yes

Is it clear how to access all supporting data?

Yes

Do you have any ethical concerns with this paper?

No

Have you any concerns about statistical analyses in this paper?

No

Recommendation?

Accept with minor revision (please list in comments)

Comments to the Author(s)

Purser et al. Depression chains in seafloor of contrasting morphology, Atacama Trench margin. A comment on Marsh et al. (2018).

This paper provides additional evidence to suggest that the ballast-collection hypothesis presented by Marsh et al. was flawed. I agree wholeheartedly with this sentiment.

I believe several reviewers (myself included) suggested that the original paper (submitted to other journals prior to its publication) should omit this material – pointing out that this was not supported by the biology of the animals themselves (negatively buoyant near the surface and more so at depth). This recommendation was not followed. I therefore very much recommend that this note is published, but suggest that the paper could say a little more (in discussion) about the lack of support for this hypothesis:

For instance, there is nothing in the ‘observations’ presented by Marsh et al. that supports ingestion of coarse material as ballast, and there is a distinct lack of support for this hypothesis within the literature. The relative paucity and variety of material ingested by deep-diving whales better supports accidental ingestion during foraging than intentional ingestion for ballast (Walker et al. 2002 documents only 2 of 20 stomachs of beaked whales containing stones, Nemoto & Nasu 1963 show various items in sperm whale stomachs). Articles referred to by Taylor (1993) suggest that gastroliths are found predominately in marine vertebrates with hydrofoil limbs (plesiosaurs, penguins, otariid pinnipeds) but not those with caudal fins (i.e. not cetaceans).

In fact, beaked whales appear to be negatively buoyant. Miller et al (2016) demonstrate negative tissue buoyancy for northern bottlenose whales (a species of beaked whale) in seawater. So it is unclear why increasing their negative buoyancy (by ingesting rocks) would be useful to them. (They already can and do change their buoyancy by altering diving lung volume.) Ingestion of ballast is not a convincing reason for whales diving 4 km to further decrease their buoyancy.

Miller, P., Narazaki, T., Isojunno, S., Aoki, K., Smout, S., and Sato, K. (2016). Body density and diving gas volume of the northern bottlenose whale (*Hyperoodon ampullatus*). *Journal of Experimental Biology* 219, 2458-2468.

Other than this, I have only minor corrections/comments:

Abstract: delete last sentence ‘and therefore impacts of this deep diving foraging behaviour on deep seafloor communities’

Results.

P2112. It would be useful to state how long these chains were (looks from figure as approx. 300m?)

P2118. ~8.5m. Perhaps worth noting that this is around a single body length of either sperm whale or Cuvier’s beaked whale.

Discussion.

P3120. Should be “observed (i.e. tagged) diving to the seafloor depths”

P3124. Reference 13 is not relevant to this statement.

P3130. Replace ‘seems’ with ‘is’ to read ‘to function in the role of gastroliths, is highly unlikely’.

Please add additional material discussed above to further substantiate this statement, e.g.

“Furthermore the paucity and variety of material ingested by deep-diving whales better supports accidental ingestion during foraging than intentional ingestion for ballast (Walker et al. documents only 2 of 20 stomachs of beaked whales containing stones). Beaked whales have negative tissue density in seawater, i.e. they are negatively buoyant (Miller et al 2016), which also suggests no benefit to ingestion of ballast.”

P3141. “Observed” in Woodside et al. (rather than ‘hypothesised’)

P3155. “quite some time” – please be more specific.

P3157. Delete “Possibly opportunistic grazing of the benthic environment is made prior a return to the surface”

References. Please check titles – should be lower case. Kenyon – should be Cuvier’s.

Figures.

Figure 1. Unclear that this figure is necessary. What is a? What is b? What are stars? Location of current study is not clear (this is not circled – it is perhaps the filled star?). Are other locations relevant – were marks observed at all of these?

Figures 2,3, and 4 should be merged into a composite single figure – and show part of fig 2. That is expanded into Fig 4 and Fig 3. (It is not apparent that Fig 4 adds much – what is green and grey part in centre?)

Figure 5. f) Unknown incirrate octopus (not octopi... there is only one in this image).

Review form: Reviewer 2 (Erin Falcone)

Is the manuscript scientifically sound in its present form?

Yes

Are the interpretations and conclusions justified by the results?

Yes

Is the language acceptable?

Yes

Is it clear how to access all supporting data?

Yes

Do you have any ethical concerns with this paper?

No

Have you any concerns about statistical analyses in this paper?

No

Recommendation?

Accept with minor revision (please list in comments)

Comments to the Author(s)

It is very exciting to see the growing body of data on deep water prey distribution, and strengthened evidence of benthic habitat use by cetaceans, that these newer survey technologies are providing. Thank you so much for taking the time to highlight this small, but important, finding. You do a good job of supporting what I think is a sound hypothesis for the formation of these depression chains while tactfully refuting the "ballast collection" hypothesis, which seemed dubious at best.

I have provided some fairly minor comments in the attached document (Appendix A). Perhaps the most significant change I would suggest relates to incorporating a more recent, and quite comprehensive, publication on dietary preferences of Cuvier's beaked whales into your discussion. I think this can improve your concluding remarks, and it also underscores the importance of your findings and the value of your survey methodology to our general understanding of this sensitive and data-deficient species.

Decision letter (RSOS-182053.R0)

23-Jan-2019

Dear Dr Purser

On behalf of the Editors, I am pleased to inform you that your Manuscript RSOS-182053 entitled "Depression chains in seafloor of contrasting morphology, Atacama Trench margin. A comment on Marsh et al. (2018) ." has been accepted for publication in Royal Society Open Science subject to minor revision in accordance with the referee suggestions. Please find the referees' comments at the end of this email.

The reviewers and handling editors have recommended publication, but also suggest some minor revisions to your manuscript. Therefore, I invite you to respond to the comments and revise your manuscript.

- Ethics statement

- Data accessibility

If you wish to submit your supporting data or code to Dryad (<http://datadryad.org/>), or modify your current submission to dryad, please use the following link:
<http://datadryad.org/submit?journalID=RSOS&manu=RSOS-182053>

- **Competing interests**

- **Authors' contributions**

- **Acknowledgements**

- **Funding statement**

Because the schedule for publication is very tight, it is a condition of publication that you submit the revised version of your manuscript before 01-Feb-2019. Please note that the revision deadline will expire at 00.00am on this date. If you do not think you will be able to meet this date please let me know immediately.

When submitting your revised manuscript, you will be able to respond to the comments made by the referees and upload a file "Response to Referees" in "Section 6 - File Upload". You can use this to document any changes you make to the original manuscript. In order to expedite the

processing of the revised manuscript, please be as specific as possible in your response to the referees. We strongly recommend uploading two versions of your revised manuscript:

on behalf of Professor Kevin Padian (Subject Editor)
 openscience@royalsociety.org

Reviewer comments to Author:

Reviewer: 1

Comments to the Author(s)

Purser et al. Depression chains in seafloor of contrasting morphology, Atacama Trench margin. A comment on Marsh et al. (2018).

This paper provides additional evidence to suggest that the ballast-collection hypothesis presented by Marsh et al. was flawed. I agree wholeheartedly with this sentiment.

I believe several reviewers (myself included) suggested that the original paper (submitted to other journals prior to its publication) should omit this material – pointing out that this was not supported by the biology of the animals themselves (negatively buoyant near the surface and more so at depth). This recommendation was not followed. I therefore very much recommend that this note is published, but suggest that the paper could say a little more (in discussion) about the lack of support for this hypothesis:

For instance, there is nothing in the ‘observations’ presented by Marsh et al. that supports ingestion of coarse material as ballast, and there is a distinct lack of support for this hypothesis within the literature. The relative paucity and variety of material ingested by deep-diving whales better supports accidental ingestion during foraging than intentional ingestion for ballast (Walker et al. 2002 documents only 2 of 20 stomachs of beaked whales containing stones, Nemoto & Nasu 1963 show various items in sperm whale stomachs). Articles referred to by Taylor (1993) suggest that gastroliths are found predominately in marine vertebrates with hydrofoil limbs (plesiosaurs, penguins, otariid pinnipeds) but not those with caudal fins (i.e. not cetaceans).

In fact, beaked whales appear to be negatively buoyant. Miller et al (2016) demonstrate negative tissue buoyancy for northern bottlenose whales (a species of beaked whale) in seawater. So it is unclear why increasing their negative buoyancy (by ingesting rocks) would be useful to them. (They already can and do change their buoyancy by altering diving lung volume.) Ingestion of ballast is not a convincing reason for whales diving 4 km to further decrease their buoyancy.

Miller, P., Narazaki, T., Isojunno, S., Aoki, K., Smout, S., and Sato, K. (2016). Body density and diving gas volume of the northern bottlenose whale (*Hyperoodon ampullatus*). *Journal of Experimental Biology* 219, 2458-2468.

Other than this, I have only minor corrections/comments:

Abstract: delete last sentence ‘and therefore impacts of this deep diving foraging behaviour on deep seafloor communities’

Results.

P2112. It would be useful to state how long these chains were (looks from figure as approx. 300m?)

P2118. ~8.5m. Perhaps worth noting that this is around a single body length of either sperm whale or Cuvier’s beaked whale.

Discussion.

P3120. Should be "observed (i.e. tagged) diving to the seafloor depths"

P3124. Reference 13 is not relevant to this statement.

P3130. Replace 'seems' with 'is' to read 'to function in the role of gastroliths, is highly unlikely'.

Please add additional material discussed above to further substantiate this statement, e.g.

"Furthermore the paucity and variety of material ingested by deep-diving whales better supports accidental ingestion during foraging than intentional ingestion for ballast (Walker et al. documents only 2 of 20 stomachs of beaked whales containing stones). Beaked whales have negative tissue density in seawater, i.e. they are negatively buoyant (Miller et al 2016), which also suggests no benefit to ingestion of ballast."

P3141. "Observed" in Woodside et al. (rather than 'hypothesised')

P3155. "quite some time" - please be more specific.

P3157. Delete "Possibly opportunistic grazing of the benthic environment is made prior a return to the surface"

References. Please check titles - should be lower case. Kenyon - should be Cuvier's.

Figures.

Figure 1. Unclear that this figure is necessary. What is a? What is b? What are stars? Location of current study is not clear (this is not circled - it is perhaps the filled star?). Are other locations relevant - were marks observed at all of these?

Figures 2,3, and 4 should be merged into a composite single figure - and show part of fig 2. That is expanded into Fig 4 and Fig 3. (It is not apparent that Fig 4 adds much - what is green and grey part in centre?)

Figure 5. f) Unknown incirrate octopus (not octopi... there is only one in this image).

Reviewer: 2

Comments to the Author(s)

It is very exciting to see the growing body of data on deep water prey distribution, and strengthened evidence of benthic habitat use by cetaceans, that these newer survey technologies are providing. Thank you so much for taking the time to highlight this small, but important, finding. You do a good job of supporting what I think is a sound hypothesis for the formation of these depression chains while tactfully refuting the "ballast collection" hypothesis, which seemed dubious at best.

I have provided some fairly minor comments in the attached document. Perhaps the most significant change I would suggest relates to incorporating a more recent, and quite comprehensive, publication on dietary preferences of Cuvier's beaked whales into your discussion. I think this can improve your concluding remarks, and it also underscores the importance of your findings and the value of your survey methodology to our general understanding of this sensitive and data-deficient species.

Author's Response to Decision Letter for (RSOS-182053.R0)

See Appendix B.

Decision letter (RSOS-182053.R1)

18-Feb-2019

Dear Dr Purser,

I am pleased to inform you that your manuscript entitled "Depression chains in seafloor of contrasting morphology, Atacama Trench margin. A comment on Marsh et al. (2018) ." is now accepted for publication in Royal Society Open Science.

on behalf of Professor Kevin Padian (Subject Editor)
openscience@royalsociety.org

Appendix A

Reviewer's comments for RSOS-182053

1. Summary
 - a. Page 1, Line 47: I would consider revising the final statement of this summary along the following lines: Given that both cruises found these depression chains in contrasting physical regions of the East Pacific, they may have a more ubiquitous distribution than just these sites. Thus, the impacts of cetacean foraging behaviour on deep seafloor communities, and the potential relevance of these prey sources to deep diving species, should be considered.
2. Introduction and Methodology
 - a. Page1, Lines 57-58: Recommend adding complete names for all acronyms upon first use (e.g. HADES-ERC and AWI OFOBS), both here and throughout the document.
3. Results
 - a. Page 2, Line 12: If "SO261/109-1" (in parentheses) is an identifier, I would add that for clarification (e.g. "survey ID SO261/109-1").
 - b. Page 2, Line 16: add a comma after [3] and change "area were" to "area was"
 - c. Page 2, Lines 27-37: Given the detailed descriptions of relevant features in figure S1 provided here, I would consider indicating these on the figure itself. I found it difficult to discern them without being more familiar with this type of imagery.
4. Discussion
 - a. Page 2, Line 54: Suggest replacing "missed" with "not detected" or "not detectable"
 - b. Page 2, Line 56: Since you use the phrase "differed greatly" to contrast the seafloor characteristics between these two surveys, were there other differences besides the presence/absence of polymetallic nodules?
 - c. Page 3, Line 41: Replace comma after [19] with semicolon
 - d. Page 3, Line 57 through end: I would rewrite this section a bit, and consider adding some discussion of findings in West et al. (2017) (below), as they pertain to the "grazing" hypothesis. They note the presence of crustaceans in stomach contents of Cuvier's beaked whales in the North Pacific and elsewhere, and while they don't indicate which (if any) of the species were benthic, it may help support your argument that whales "graze" on benthic organisms when the opportunity presents itself. Given the energetic

investment in commuting to such depths to feed, it does make sense that whales would take advantage of these low-cost predation opportunities, especially if they encounter lower densities of more desirable prey on a given dive. Also, while benthic prey may not form a significant portion of the typical diet, they may be of higher importance for some whales (*e.g.* younger animals that may be less successful at foraging on more active targets, individuals that are stressed or physiologically compromised in some way, or whales experiencing acoustic disturbances that impede their ability to pursue more active prey species). I think this would both strengthen your basic hypothesis of the etiology of depression chains, and also increase the relevance of your paper to the field of deep-water cetacean ecology.

- i. West KL, Walker WA, Baird RW, Mead JG, Collins PW (2017) Diet of Cuvier's beaked whales *Ziphius cavirostris* from the North Pacific and a comparison with their diet world-wide. *Mar Ecol Prog Ser* 574:227-242.
<https://doi.org/10.3354/meps12214>

5. Figures

- a. Figure Caption 1: "a" and "b" are not indicated in the panel. A clearer explanation of what the stars represent would be helpful (*e.g.* "stars mark approximate locations of OFOBS deployments"). The caption suggests there should be a circled deployment, but there isn't one- I assume this should actually be the black star?
- b. Figure Caption 2: There should be a period after (2018).
- c. Figure caption 3: I would revise the first sentence to read "Image of a depression feature taken from 1.5 m altitude".
- d. Figure caption 4: This caption is a little confusing, but I don't understand it well enough to suggest a revision.
- e. Figure caption 5: correct "sp.." to "spp.", add ")" after "b", replace comma with period after "mound" for consistency with other formatting.

Alfred-Wegener-Institut, Postfach 12 01 61, 27515 Bremerhaven

Appendix B

February 7th 2019

RE: Comment submission – Manuscript ID RSOS-182053

Dear RSOS team,

Thank you for the swift review of the initial draft of our comment manuscript, particularly over the festive period!

We were greatly pleased to receive such a positive review to the initial draft, supported by really excellent and useful suggestions by the reviewers. The great majority of these have been integrated into the new version, as can be seen in the attached 'track changes' version of the manuscript. To further assist in this, we have provided a detailed response to each of the reviewer comments on the following pages of this letter.

We would like to thank again you and the reviewers for the time taken to consider this submission, and particularly the care taken by those reviewers in providing constructive suggestions.

Yours sincerely,

Autun Purser (on behalf of all authors).

Dr Autun Purser

autun.purser@awi.de

Alfred-Wegener-Institut
Helmholtz-Zentrum für
Polar- und Meeresforschung
BREMERHAVEN

Am Handelshafen 12
27570 Bremerhaven
Telefon 0471 4831-0
Telefax 0471 4831-1149
www.awi.de

Stiftung des öffentlichen Rechts

Sitz der Stiftung:
Am Handelshafen 12
27570 Bremerhaven
Telefon 0471 4831-0
Telefax 0471 4831-1149
www.awi.de

Vorsitzender des Kuratoriums:
MinDir Dr. Karl Eugen Huthmacher
Direktorium:
Prof. Dr. Dr. h.c. Karin Lochte
(Direktorin)
Dr. Karsten Wurr
(Verwaltungsdirektor)
Dr. Uwe Nixdorf
(Stellvertretender Direktor)
Prof. Dr. Karen H. Wiltshire
(Stellvertretende Direktorin)

Bankverbindung:
Commerzbank AG,
Bremerhaven
BIC/Swift COBADEFF292
IBAN DE12292400240349192500
UST-Id-Nr. DE 114707273

DETAILED RESPONSE TO REVIEWER COMMENTS:
REVIEWER TEXT IN BLACK, **OUR RESPONSE GIVEN IN RED**

Reviewer: 1

Comments to the Author(s)

Purser et al. Depression chains in seafloor of contrasting morphology, Atacama Trench margin. A comment on Marsh et al. (2018).

COMMENT: This paper provides additional evidence to suggest that the ballast-collection hypothesis presented by Marsh et al. was flawed. I agree wholeheartedly with this sentiment.

I believe several reviewers (myself included) suggested that the original paper (submitted to other journals prior to its publication) should omit this material – pointing out that this was not supported by the biology of the animals themselves (negatively buoyant near the surface and more so at depth). This recommendation was not followed. I therefore very much recommend that this note is published, but suggest that the paper could say a little more (in discussion) about the lack of support for this hypothesis:

For instance, there is nothing in the ‘observations’ presented by Marsh et al. that supports ingestion of coarse material as ballast, and there is a distinct lack of support for this hypothesis within the literature. The relative paucity and variety of material ingested by deep-diving whales better supports accidental ingestion during foraging than intentional ingestion for ballast (Walker et al. 2002 documents only 2 of 20 stomachs of beaked whales containing stones, Nemoto & Nasu 1963 show various items in sperm whale stomachs). Articles referred to by Taylor (1993) suggest that gastroliths are found predominately in marine vertebrates with hydrofoil limbs (plesiosaurs, penguins, otariid pinnipeds) but not those with caudal fins (i.e. not cetaceans).

RESPONSE: These useful comments have now all been incorporated into the manuscript, and appropriately referenced, such as in the last paragraph of the discussion. We favour the idea that these stones, when they have been found, are accidental results of the benthic opportunistic feeding, and we believe we have made this opinion clearer in this draft.

In fact, beaked whales appear to be negatively buoyant. Miller et al (2016) demonstrate negative tissue buoyancy for northern bottlenose whales (a species of beaked whale) in seawater. So it is unclear why increasing their negative buoyancy (by ingesting rocks) would be useful to them. (They already can and do change their buoyancy by altering diving lung volume.) Ingestion of ballast is not a convincing reason for whales diving 4 km to further decrease their buoyancy.

Miller, P., Narazaki, T., Isojunno, S., Aoki, K., Smout, S., and Sato, K. (2016). Body density and diving gas volume of the northern bottlenose whale (*Hyperoodon ampullatus*). *Journal of Experimental Biology* 219, 2458-2468.

RESPONSE: This comment, and the associated reference, now form the initial section of paragraph 4 of the discussion.

Other than this, I have only minor corrections/comments:

COMMENT: Abstract: delete last sentence 'and therefore impacts of this deep diving foraging behaviour on deep seafloor communities'

RESPONSE: The last sentence of the abstract has been rewritten.

Results.

COMMENT: P2I12. It would be useful to state how long these chains were (looks from figure as approx. 300m?)

RESPONSE: The lengths of the chains is now discussed in the first paragraph of the results in more detail.. up to 500 m in our data.

COMMENT: P2I18. ~8.5m. Perhaps worth noting that this is around a single body length of either sperm whale or Cuvier's beaked whale.

RESPONSE: The comparison of this depression spacing to a beaked whale length is now added in the first paragraph of the results.

Discussion.

COMMENT: P3I20. Should be "observed (i.e. tagged) diving to the seafloor depths"

RESPONSE: This change has been made as suggested.

COMMENT: P3I24. Reference 13 is not relevant to this statement.

RESPONSE: Removed as suggested.

COMMENT: P3I30. Replace 'seems' with 'is' to read 'to function in the role of gastroliths, is highly unlikely'. Please add additional material discussed above to further substantiate this statement, e.g.

"Furthermore the paucity and variety of material ingested by deep-diving whales better supports accidental ingestion during foraging than intentional ingestion for ballast (Walker et al. documents only 2 of 20 stomachs of beaked whales containing stones). Beaked whales have negative tissue density in seawater, i.e. they are negatively buoyant (Miller et al 2016), which also suggests no benefit to ingestion of ballast."

RESPONSE: The wording 'seems' has been changed as suggested. Our opinion that where stones have been consumed, it is as accidental consumption, rather than an intended result of deep diving, has been made clearer in the revised draft.

COMMENT: P3I41. "Observed" in Woodside et al. (rather than 'hypothesised')

RESPONSE: This change has been made as suggested.

COMMENT: P3I55. "quite some time" – please be more specific.

RESPONSE: This embarrassingly unspecific timeframe has been changed to ~50 mins, as shown in the figures from the cited paper.

COMMENT: P3I57. Delete “Possibly opportunistic grazing of the benthic environment is made prior a return to the surface”

RESPONSE: This sentence has been changed to match the suggestions of the other reviewer, giving more info on why the whales may in some instances be utilising these benthic food sources.

COMMENT: References. Please check titles – should be lower case. Kenyon – should be Cuvier’s.

RESPONSE: Cuvier to Cuvier’s has been corrected. As to the other formatting issues, we used the RSOS template here in the reference managing software, so I presume all is now correct for the journal?

Figures.

COMMENT: Figure 1. Unclear that this figure is necessary. What is a? What is b? What are stars? Location of current study is not clear (this is not circled – it is perhaps the filled star?). Are other locations relevant – were marks observed at all of these?

RESPONSE: We have modified the legend to this figure to address the star issue, and also to comment that at only one of these sites sidescan data was collected with our device. We wanted to place our location clearly geographically, as it is coincident with historical reports of whale observations and interactions, and also shows what may be an interesting relationship with the trench flanks, should future work show a preference for such activities in such areas. We are also happy to remove if necessary.

COMMENT: Figures 2,3, and 4 should be merged into a composite single figure – and show part of fig 2. That is expanded into Fig 4 and Fig 3. (It is not apparent that Fig 4 adds much – what is green and grey part in centre?)

RESPONSE: We have improved the legends of Figs 3 and particularly 4. The reason we did not combine into on composite image is that we wanted to upload these figures in near native resolution, as the interested reader can actually get quite a lot of information on fauna and topography by zooming in on these, if they are interested to do so. If we made a composite image, some of this data would be lost.

COMMENT: Figure 5. f) Unknown incirrate octopus (not octopi... there is only one in this image).

RESPONSE: This change has been made as suggested.

Reviewer: 2

Comments to the Author(s)

COMMENT: It is very exciting to see the growing body of data on deep water prey distribution, and strengthened evidence of benthic habitat use by cetaceans, that these newer survey technologies are providing. Thank you

so much for taking the time to highlight this small, but important, finding. You do a good job of supporting what I think is a sound hypothesis for the formation of these depression chains while tactfully refuting the "ballast collection" hypothesis, which seemed dubious at best.

RESPONSE: We believe that the depression chains are sufficiently robust in our sidescan data to draw further attention to them as indicators of major seafloor interactions. We additionally believe that the general readers have jumped on the whale / nodule association with undue vigour following the publication of the original paper – despite this being only part of the story in that paper. Given the sensitivity of the nodule mining topic, we consider it paramount our observations of the same sort of depression chains in nodule free areas are published ASAP before the idea that deep diving whales ‘need’ nodule fields becomes an accepted paradigm.

I have provided some fairly minor comments in the attached document. Perhaps the most significant change I would suggest relates to incorporating a more recent, and quite comprehensive, publication on dietary preferences of Cuvier's beaked whales into your discussion. I think this can improve your concluding remarks, and it also underscores the importance of your findings and the value of your survey methodology to our general understanding of this sensitive and data-deficient species.

Reviewer's comments for RSOS-182053 – imported from review .pdf:

1. Summary

COMMENT: a. Page 1, Line 47: I would consider revising the final statement of this summary along the following lines: Given that both cruises found these depression chains in contrasting physical regions of the East Pacific, they may have a more ubiquitous distribution than just these sites. Thus, the impacts of cetacean foraging behaviour on deep seafloor communities, and the potential relevance of these prey sources to deep diving species, should be considered.

RESPONSE: This sentence has been reworded to better integrate the comments of both reviewers.

2. Introduction and Methodology

COMMENT: a. Page1, Lines 57-58: Recommend adding complete names for all acronyms upon first use (e.g. HADES-ERC and AWI OFOBS), both here and throughout the document.

RESPONSE: This has been done... though HADES itself just refers to the oceanic depth at which the majority of the project work is conducted, rather than being an acronym.

3. Results

COMMENT: a. Page 2, Line 12: If “SO261/109-1” (in parentheses) is an identifier, I would add that for clarification (e.g. “survey ID SO261/109-1”).

RESPONSE: These references to the particular survey have been changed throughout the document as suggested.

COMMENT: b. Page 2, Line 16: add a comma after [3] and change “area were” to “area was”

RESPONSE: This has been done as suggested.

COMMENT: c. Page 2, Lines 27-37: Given the detailed descriptions of relevant features in figure S1 provided here, I would consider indicating these on the figure itself. I found it difficult to discern them without being more familiar with this type of imagery.

RESPONSE: This has been done as suggested.

4. Discussion

COMMENT: a. Page 2, Line 54: Suggest replacing “missed” with “not detected” or “not detectable”

RESPONSE: This has been done as suggested.

COMMENT: b. Page 2, Line 56: Since you use the phrase “differed greatly” to contrast the seafloor characteristics between these two surveys, were there other differences besides the presence/absence of polymetallic nodules?

RESPONSE: The environmental characteristics of the two sites are now described better, with the abundance of hard substrate at one, and the absence at the other, now presented more clearly in the text.

COMMENT: c. Page 3, Line 41: Replace comma after [19] with semicolon

RESPONSE: This has been done as suggested.

COMMENT: d. Page 3, Line 57 through end: I would rewrite this section a bit, and consider adding some discussion of findings in West et al. (2017) (below), as they pertain to the “grazing” hypothesis. They note the presence of crustaceans in stomach contents of Cuvier’s beaked whales in the North Pacific and elsewhere, and while they don’t indicate which (if any) of the species were benthic, it may help support your argument that whales “graze” on benthic organisms when the opportunity presents itself. Given the energetic investment in commuting to such depths to feed, it does make sense that whales would take advantage of these low-cost predation opportunities, especially if they encounter lower densities of more desirable prey on a given dive. Also, while benthic prey may not form a significant portion of the typical diet, they may be of higher importance for some whales (e.g. younger animals that may be less successful at foraging on more active targets, individuals that are stressed or physiologically compromised in some way, or whales experiencing acoustic disturbances that impede their ability to pursue more active prey species). I think this would both strengthen your basic hypothesis of the etiology of depression chains, and also increase the relevance of your paper to the field of deep-water cetacean ecology.

West KL, Walker WA, Baird RW, Mead JG, Collins PW (2017) Diet of Cuvier's beaked whales *Ziphius cavirostris* from the North Pacific and a comparison with their diet world-wide. *Mar Ecol Prog Ser* 574:227-242.

<https://doi.org/10.3354/meps12214>

RESPONSE: These useful observations are now incorporated into the discussion, along with the recent reference.

5. Figures

COMMENT: a. Figure Caption 1: "a" and "b" are not indicated in the panel. A clearer explanation of what the stars represent would be helpful (e.g. "stars mark approximate locations of OFOBS deployments"). The caption suggests there should be a circled deployment, but there isn't one- I assume this should actually be the black star?

RESPONSE: This figure caption is improved.

COMMENT: b. Figure Caption 2: There should be a period after (2018).

RESPONSE: This has been done as suggested.

COMMENT: c. Figure caption 3: I would revise the first sentence to read "Image of a depression feature taken from 1.5 m altitude".

RESPONSE: This has been done as suggested.

COMMENT: d. Figure caption 4: This caption is a little confusing, but I don't understand it well enough to suggest a revision.

RESPONSE: This figure caption has been wholly rewritten for clarity.

COMMENT:e. Figure caption 5: correct "sp." to "spp.", add ")" after "b", replace comma with period after "mound" for consistency with other formatting.

RESPONSE: This has been done as suggested.